# Response to Initial Anti-Vascular Endothelial Growth Factor for Diabetic Macular Edema Is Significantly Correlated with Response to Third Consecutive Monthly Injection

**DOI:** 10.3390/jcm11216416

**Published:** 2022-10-29

**Authors:** Satoshi Maeda, Masahiko Sugimoto, Yumiho Tenma, Hideyuki Tsukitome, Kumiko Kato, Shinichiro Chujo, Yoshitsugu Matsui, Hisashi Matsubara, Mineo Kondo

**Affiliations:** Department of Ophthalmology, Mie University Graduate School of Medicine, 2-174, Edobashi, Tsu 514-8507, Japan

**Keywords:** anti-vascular epithelial growth factor, consecutive monthly injections, diabetic macular edema, best-corrected visual acuity, central macular thickness

## Abstract

Purpose: To determine whether the response to the initial anti-vascular endothelial growth factor (anti-VEGF) injection to treat diabetic macular edema (DME) is significantly correlated with the response to the third consecutive monthly injection of the same anti-VEGF agent. Methods: Seventy eyes with DME that were treated with an anti-VEGF agent (16 eyes with 1.25-mg bevacizumab, 35 eyes with 0.5-mg ranibizumab, and 19 eyes with 2.0-mg aflibercept) were studied. They were treated with three consecutive monthly injections of one of the three anti-VEGF agents. The best-corrected visual acuity (BCVA) in the logarithm of the minimum angle of resolution (logMAR units) and the central macular thickness (CMT) were measured at the baseline, 1 week after the initial injection, and 1 month after the third consecutive monthly injection. The changes of both values from the baseline 1 week after the initial injection (day 7) and 1 month after the third monthly injection were determined. The significance of the correlations between the BCVA and the CMT was determined. Results: The mean BCVA improved significantly for all three agents (0.38 ± 0.22 logMAR units at the baseline to 0.27 ± 0.25 logMAR units) after the three monthly injections (*p* < 0.05, repeated ANOVA). For all cases, a moderate but significant correlation was found between the BCVA at day 7 and 1 month after the third injection (r = 0.58, *p* < 0.01; Spearman’s rank correlation). No significant correlation was found for bevacizumab (r = 0.09, *p* = 0.73), moderate correlation was found for ranibizumab (r = 0.42, *p* < 0.05), and a strong correlation was found for aflibercept (r = 0.83, *p* < 0.001) between the BCVA at day 7 and at 1 month after the third injection. The mean CMT improved significantly for all three agents (481.9 ± 96.3 μm at the baseline to 364.1 ± 116.0 μm after the three monthly injections, *p* < 0.05), and a moderate correlation was found for the three agents between CMT at day 7 to that at one month after the third anti-VEGF injection (r = 0.54, *p* < 0.01). A moderate correlation was found for all three agents between CMT at day 7 to that at one month after the third anti-VEGF injection (r = 0.68 for bevacizumab, r = 0.41 for ranibizumab and r = 0.53 for aflibercept, *p* < 0.05). Conclusions: The significant correlations between the results on day 7 to that one month after the third anti-VEGF treatment for DME indicates that the long-term effects of anti-VEGF therapy can be predicted by the short-term response. In addition, the results indicate that there may be differences in the effectiveness between the three anti-VEGF agents.

## 1. Introduction

Diabetic macular edema (DME) is a complication of diabetic retinopathy, and it is accompanied by a reduction in vision [1]. It is well established that intravitreal injections of anti-vascular endothelial growth factor (anti-VEGF) agents are effective in resolving edema, and anti-VEGF therapy has become the first-line therapy. At present, three anti-VEGF agents are being used: bevacizumab (Avastin, Genentech), ranibizumab (Lucentis, Genentech), and aflibercept (Eylea, Regeneron Pharmaceuticals) [2,3,4]. Of special note, bevacizumab was the only agent that could be used in Japan before 2014 because no other anti-VEGF agents were approved. Although there has not been an effective treatment for DME that can improve visual function, the Diabetic Retinopathy Clinical Research Network (DRCR.net) protocol of 2 years of continuous monthly treatment has been reported to be effective with 60% of the patients showing a reduction in the retinal thickness. However, there is a problem in that the other 40% of patients remained nonresponders [4]. Similar findings were reported in another study, even in eyes that received multiple injections including a loading phase [5]. Another problem of anti-VEGF therapy is its high cost and the need for multiple injections [6].

In addition to anti-VEGF therapy, there are other treatment options for DME, such as intravitreal steroids, pars plana vitrectomy (PPV) for patients with vitreomacular traction, and focal laser photocoagulation. It is generally accepted that after a certain number of anti-VEGF injections is found not to be effective, the anti-VEGF agents should be discontinued and the treatment switched to these alternative therapies. Thus, it is important to identify these nonresponders early so that alternative treatments can be started. Unfortunately, there has not been a study performed to identify the nonresponders early.

Thus, this study aimed to determine whether the response to the initial anti-VEGF therapy for DME is significantly correlated with the response at one month after the third consecutive monthly injection.

## 2. Patients and Methods

This was a retrospective, observational, single-center study. The study protocol conformed to the tenets of the Declaration of Helsinki for research involving human subjects. The Institutional Review Board of Mie University Graduate School of Medicine (No.1736) approved the protocol which conformed to the principles of Good Clinical Practice and the Helsinki guidelines. We registered this prospective study at http://www.umin.ac.jp (accessed on 25 October 2022, UMIN ID 000012094). Because this was a retrospective study, patients were not matched or blinded.

### 2.1. Inclusion and Exclusion Criteria

Patients were treated with bevacizumab, ranibizumab, or aflibercept from January 2012 to December 2016 at Mie University Hospital. Because we had stopped medical examinations after 2017 due to the overcrowding of the clinic, this term was defined. The inclusion criteria were: patients ≥20 years of age with type I or type II diabetes (this is because the safety of anti-VEGF agents was not established in 2012), presence of abnormal retinal vessels, best-corrected visual acuity (BCVA) of ≥20/320, DME involving the fovea, and central macular thickness (CMT) ≥300 μm measured as the mean retinal thickness in the central 1 mm-diameter circle by optical coherence tomography (OCT). The CMT of 300 μm was as defined in the RESTORE study [7]. The exclusion criteria were any retinal photocoagulation treatment in the study eye within the 3 months preceding the initial anti-VEGF injection, eyes with an ischemic macular region involving the fovea, presence of other ocular diseases causing vision reduction, such as age-related macular degeneration and severe proliferative DR, signs of optic nerve atrophy, glaucoma or intraocular pressure of more than 24 mmHg, prior vitreous surgery, aphakia, anti-VEGF treatment on either eye within 3 months preceding the initial injection, cloudy optic media including cataracts through which high-quality fundus photographs or OCT images could not be obtained, a history of cataract surgery in the study eye within the previous 3 months, a history of cerebrovascular accident, myocardial infarction or other systemic disease requiring medications that could affect the results, severe renal failure with a creatinine level of 2.0 mg/dL or more or worse than stage IIb nephropathy as defined by the classification of diabetic nephropathy, poorly controlled hypertension with a systolic BP higher than 200 mmHg or a diastolic BP higher than 110 mmHg, poorly controlled diabetes mellitus with a hemoglobin A1c level of 12.0% or more, and patients who were judged to be ineligible for any other reason by the investigators. 

### 2.2. Procedures

The patients were treated with three monthly injections of one of the three anti-VEGF agents. Each patient received a comprehensive ophthalmological examination before the initial anti-VEGF injection that included measurements of the BCVA, slit-lamp biomicroscopy of the anterior segment, indirect ophthalmoscopy of the posterior pole, and spectral-domain OCT imaging of the macular area. The BCVA and CMT were measured at the baseline, 1 week after the initial injection, and 1 month after the third consecutive monthly injection (Figure 1). The changes in the BCVA and CMT from the baseline to 1 week after the initial injection (day 7) and 1 month after the third consecutive monthly injection were determined. The correlation between the BCVA and CMT on day 7 and 1 month after the third monthly injection of the anti-VEGF agent was determined. We also evaluated the differences in the effectiveness of the three anti-VEGF agents.

The changes of the best-corrected visual acuity (BCVA) and central macular thickness (CMT) from baseline to 1 week after the initial anti-vascular endothelial growth factor (anti-VEGF) injection were defined as occurring on day 7. The changes of both values from the baseline 1 month after the third consecutive monthly injection were defined as one month after the third anti-VEGF injection. The correlations between day 7 and month 1 of the BCVA and CMT values were determined.

### 2.3. Measurement of BCVA and CMT

The BCVA was measured with a Landolt chart at every visit. We converted the decimal BCVA to the logarithm of the minimum angle of resolution (logMAR) units for the statistical analyses. We determined the degree of DME from the images recorded using a Heidelberg Spectralis OCT instrument (Heidelberg Engineering Inc., Heidelberg, Germany). Using the bundled software, we measured the CMT as the thickness between the internal limiting membrane and the retinal pigment epithelium at the center of a 1 mm-diameter circle of the ETDRS thickness map.

### 2.4. Intravitreal Anti-VEGF Agent Injections

The anti-VEGF agents were injected under topical anesthesia. Each patient received 0.05 mL of the same anti-VEGF agent (1.25-mg bevacizumab, 0.5-mg ranibizumab, or 2.0-mg aflibercept) which was injected intravitreally with a 30-gauge needle inserted 4 mm posterior to the corneal limbus under sterile conditions. After the injection, all of the patients received topical levofloxacin hydrate (1.5% Cravit ophthalmic solution) for 3 days. We performed three consecutive monthly injections of the same anti-VEGF agent.

### 2.5. Statistical Analyses

The results are presented as the means ± standard deviations (SDs). We used d-values that are defined as the rate of change from the baseline values. The d-values or reduction rates were calculated as follows:***d-values (%) = 100 − (value after injection/value before injection × 100).***(1)

The significance of the differences in the data was determined by a 2-way repeated analysis of variance followed by Bonferroni post hoc tests for the comparisons of the means. Spearman’s rank-order correlation coefficient was used to determine the significance of the correlations among the variables. The strength of the correlations (r-value) was designated as: 0.0 to 0.1, not correlated or very weakly correlated; 0.1 to 0.3, weak or low; 0.3 to 0.5, moderate; 0.5 to 0.9, strong or high. The results were statistically significant when the *p*-value was <0.05. The statistical evaluations were performed with the Statcel 4 Statistical Program (Statcel; OMC, Saitama, Japan).

## 3. Results

The baseline demographics of the patients are shown in Table 1. Seventy eyes of 70 patients with DME were studied. 

The average age of the patients was 63.6 ± 11.1 years with a range from 26 to 80 years. Of the 70 eyes, 16 eyes were treated with 1.25 mg bevacizumab (81.8% of treatment naïve patients), 35 eyes with 0.5 mg ranibizumab (81.0% of treatment naïve patients), and 19 eyes with 2.0 mg aflibercept (77.3% of treatment naïve patients). The age differences, BCVA, and CMT at the baseline among the groups treated with the different anti-VEGF agents were not significant.

### 3.1. BCVA and CMT Improved Significantly for All Agents (Figure 2)

The mean BCVA improved significantly for all three agents 1 month after the third monthly injection of the anti-VEGF agents. For all eyes, the BCVA was 0.38 ± 0.22 logMAR units at the baseline and 0.27 ± 0.25 logMAR units one month after the third monthly injection (*p* < 0.05, repeated ANOVA). The mean CMT improved significantly for all three agents. For all eyes, the CMT was 481.9 ± 96.3 μm at the baseline and 364.1 ± 116.0 μm at 1 month after the third monthly injection (*p* < 0.05). If we defined CMT > 350 μm as recurrence, recurrence was 62.5% (all eyes), 54.5% (bevacizumab), 54.8% (ranibizumab) and 81.8% (aflibercept) for d7, and 54.9% (all eyes), 52.4% (bevacizumab), 38.1% (ranibizumab) and 52.6% (aflibercept) for dL.

**Figure 2 jcm-11-06416-f002:**
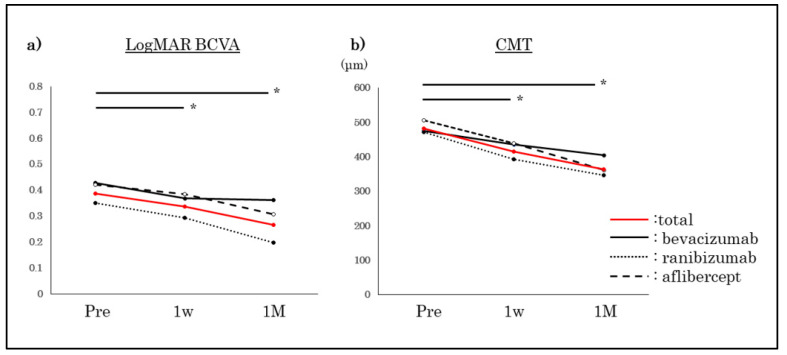
Changes of BCVA (**a**) and CMT(**b**) after anti-VEGF injections of three VEGF agents. The mean BCVA and CMT improved significantly after the intravitreal injection of each of the three anti-VEGF agents (total, bevacizumab, ranibizumab, and aflibercept). The data are the means ± SDs. *: *p* < 0.05, repeated ANOVA.

### 3.2. Correlations between BCVA and CMT at Day 7 and 1 Month after the Third Monthly Injection of Anti-VEGF Agent(Table 2)

For all cases, a strong correlation was found between the BCVA on day 7 after the initial anti-VEGF treatment and that one month after the third injection (r = 0.58, *p* < 0.01; Spearman correlation; Figure 3) and between the CMT on day 7 and the CMT at one month after the third anti-VEGF injection (r = 0.54, *p* < 0.01; Figure 3). No significant correlation was found for bevacizumab (r = 0.09, *p* = 0.73), a strong correlation was found for ranibizumab (r = 0.42, *p* < 0.05), and aflibercept (r = 0.83, *p* < 0.001) between BCVA at day 7 after the initial anti-VEGF treatment and the BCVA at one month after the third anti-VEGF injection. Correlation was found for all three agents between the CMT on day 7 to that one month after the third anti-VEGF treatment (r = 0.68 for bevacizumab; r = 0.41 for ranibizumab; and r = 0.53 for aflibercept, all *p* < 0.05).

**Table 2 jcm-11-06416-t002:** Correlation between d7 value and dL value for BCVA and CMT.

(r-Value)	Total	Bevacizumab	Ranibizumab	Aflibercept
BCVA (logMAR)	0.58 **	0.09	0.42 *	0.83 **
CMT	0.54 **	0.68 *	0.41 **	0.53 **

BCVA, best-corrected visual acuity; CMT, central macular thickness; logMAR, logarithm of the minimum angle of resolution. *: *p* < 0.05, **: *p* < 0.01, Spearman’s rank-order correlation coefficient.

## 4. Discussion

The advent of anti-VEGF treatment has revolutionized the treatment of DME. It is believed that even cases with poor vision can be improved by increasing the frequency of the anti-VEGF injections and continuous treatment is recommended in poor responders [8]. However, even if the response to anti-VEGF agents is not as strong as expected after multiple injections, the time and expense involved in the treatments are problems. Thus, the identification of a marker for eyes that are poor responders to anti-VEGF agents at an early stage will be of great help.

Our results showed that the response at 7 days after the first injection of an anti-VEGF agent was significantly correlated with the response at one month after the third anti-VEGF injection. Thus, our results indicated that the therapeutic effects can be predicted after the results of the initial treatment. Similarly, Gonzalez reported that the BCVA at 12 weeks after a ranibizumab injection for DME was significantly associated with the outcome at 3 years [9]. In addition, two other studies have reported that the short-term effects (1 day to 3 months) can predict the therapeutic outcome of the anti-VEGF treatment for DME [10,11]. Thus, the earlier prediction is desirable from a patient’s burden, and it would be ideal if the therapeutic effects could be predicted only by the response to the first injection.

We found that the results of the initial ranibizumab and aflibercept injections were associated with an improvement in the BCVA. This indicated that the short-term effects on the BCVA can predict the BCVA after continuous injections and it may be appropriate to consider adjustments to the treatment regimen or switch to other therapy if the short-term response is not good. This will be of benefit to the patients by reducing the time and financial burdens.

We also found that the response of the BCVA to the three anti-VEGF agents was different. The DRCR.net protocol T study reported that aflibercept was more effective in patients with poorer initial BCVA and inferiority of bevacizumab compared with the other two anti-VEGF agents. We found a superiority of ranibizumab and aflibercept on the BCVA. Thus, there may exist differences among these agents in the short-term response in different ocular parameters, and bevacizumab may not be as efficient as the other anti-VEGF agents.

Our study has some limitations. First, our study had a small sample size, and only the short-term results were analyzed. It is not clear whether our result could also reflect long-term effectiveness. We calculated the sample size needed for power with an effect size of 0.3, α-error of 0.05, and power of 0.95, and found that the total sample size was estimated to be 34 for each group. Although there is a possibility of a small sample size for our study, previous similar studies also reported that short-term effects can predict the therapeutic outcome of the anti-VEGF treatment of DME with a small number of fewer than 20 eyes [10,11]. Therefore, we believe that our results are reliable for determining short-term effects. Although we found small differences in the effectiveness between the three agents, this was a retrospective study, and the number of enrolled patients was relatively small and not matched. Therefore, we need further investigations on a large number of patients with a prospective and well-matched design. Second, there were significant correlations between the day 7 value and that at one month after the third injection value for BCVA with ranibizumab and aflibercept but not with bevacizumab. These differences among the three agents were found but there may have been a selection bias for the patients studied because this was a retrospective study. In addition, the d-values of the CMT were significantly correlated with bevacizumab. Finally, there exist patients whose CMT remains over 350 μm, generally defined as a nonresponder. Similar to previous randomized controlled trials, there exists nonresponder (low d7 or dL) and responder (high d7 or dL), and we did not compare the findings between these two groups. We need to further consider these limitations.

## 5. Conclusions

In conclusion, the short-term effects of the anti-VEGF treatment of DME can predict treatment effectiveness after three consecutive monthly injections. In addition, our results showed that there may be differences in the responses to the three anti-VEGF agents.

## Figures and Tables

**Figure 1 jcm-11-06416-f001:**
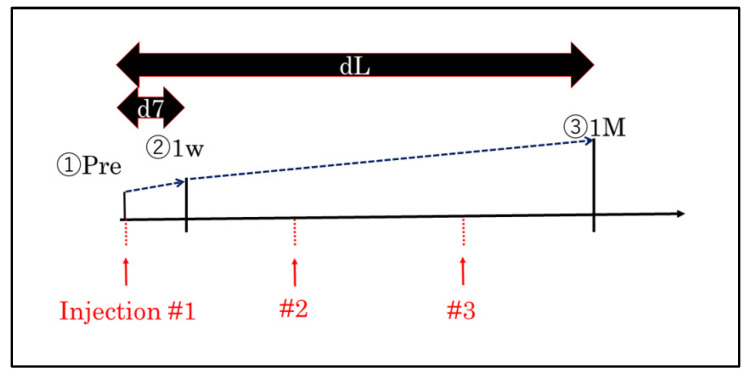
Flow chart of this study.

**Figure 3 jcm-11-06416-f003:**
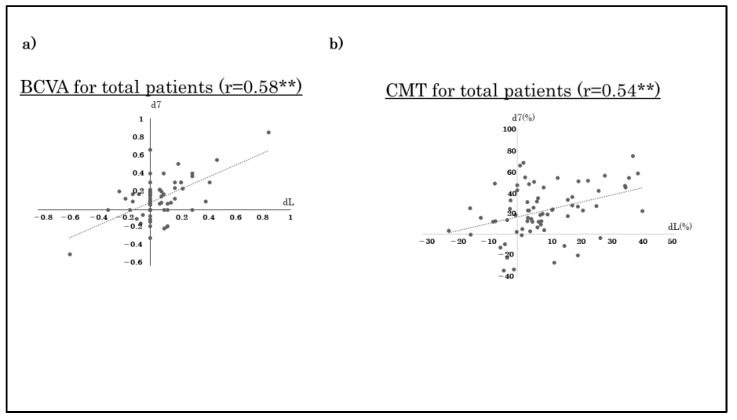
Correlations between the values at day 7 and month 1 after the third anti-VEGF injection. A moderate correlation can be seen between the BCVA at day 7 and one month after the third injection for all cases (**a**, r = 0.58; *p* < 0.01; Pearson correlation coefficient). A moderate correlation can be seen between the CMT on day 7 and one month after the third anti-VEGF injection (**b**, r = 0.54; *p* < 0.01). **: *p* < 0.01, Spearman’s rank-order correlation coefficient.

**Table 1 jcm-11-06416-t001:** Baseline demographics of patients.

	Age (yrs)	Pre-BCVA (logMAR)	Pre-CMT (μm)
Total (n = 70)	63.6 ± 11.1	0.38 ± 0.22	481.9 ± 96.3
Bevacizumab (n = 16)	60.1 ± 15.2	0.43 ± 0.29	475.6 ± 90.4
Ranibizumab (n = 35)	64.2 ± 10.0	0.35 ± 0.19	471.9 ± 107.7
Aflibercept (n = 19)	65.5 ± 8.7	0.42 ± 0.29	505.7 ± 77.2

Data are shown as mean ± standard deviation. BCVA, best-corrected visual acuity; CMT, central macular thickness; logMAR, logarithm of the minimum angle of resolution.

## Data Availability

The datasets used during the current study are available from the corresponding author on request.

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
