# Peer review of "Response to Initial Anti-Vascular Endothelial Growth Factor for Diabetic Macular Edema Is Significantly Correlated with Response to Third Consecutive Monthly Injection"

_jcm, 2022, doi:10.3390/jcm11216416_

Round 1

Reviewer 1 Report

In their study, the authors investigate possible correlation between early and later response to anti-VEGF treatment in DME. The study is well written and well conducted, and there are some comments that should be addressed.

Introduction

It should be mentioned that bevacizumab is used off-label. Especially considering that the results indicate that bevacizumab is not as effective as its approved counterparts, this is worth mentioning.

Pars plana vitrectomy for DME without vitreomacular traction is actually controversially discussed in the literature. Maybe that would be worth mentioning, too.

Methods

Please indicate how the patients were selected to be treated with the specific VEGF inhibitors. As it is a retrospective study, I do suspect that it was not “matched” or blinded, please mention this in the text.

Please indicate the clinic in which the treatment was performed.

Results

Figure 2

Please indicate in legend, what exactly is significantly different (I take it is the “total” difference, but as there are 4 different lines shown, please indicate which one is meant)

Discussion

General, the authors mention that all patients benefitted from the treatment. In the introduction, they describe that there are 40% non-responders. Please discuss, why there were no non-responders among the study subjects and whether this could be a problem for extrapolating the results. Maybe non-responders show a different correlation?

The authors briefly discuss that bevacizumab may not be as efficient as the other VEGF inhibitors. As mentioned above, bevacizumab is not approved for the use in intravitreal application, this should also be mentioned in the discussion.

Author Response

Reply to Reviewer 1

1) It should be mentioned that bevacizumab is used off-label. Especially considering that the results indicate that bevacizumab is not as effective as its approved counterparts, this is worth mentioning.

A) We add explanation about this at L48.

2) Pars plana vitrectomy for DME without vitreomacular traction is actually controversially discussed in the literature. Maybe that would be worth mentioning, too.

A) We add explanation about this at L60.

3) Please indicate how the patients were selected to be treated with the specific VEGF inhibitors. As it is a retrospective study, I do suspect that it was not “matched” or blinded, please mention this in the text.

A) We add explanation about this at L77.

4) Please indicate the clinic in which the treatment was performed.

A) We add explanation about this at L82.

5) Figure 2. Please indicate in legend, what exactly is significantly different (I take it is the “total” difference, but as there are 4 different lines shown, please indicate which one is meant).

A) Significant differences were observed for total and all of three anti-VEGF agents (bevacizumab, ranibizumab and aflibercept). We add explanation about this at L190 (Figure 2 legend).

6) General, the authors mention that all patients benefitted from the treatment. In the introduction, they describe that there are 40% non-responders. Please discuss, why there were no non-responders among the study subjects and whether this could be a problem for extrapolating the results. Maybe non-responders show a different correlation?

A) We add ratio of non-responder (CMT>350μm) at L181.

7) The authors briefly discuss that bevacizumab may not be as efficient as the other VEGF inhibitors. As mentioned above, bevacizumab is not approved for the use in intravitreal application, this should also be mentioned in the discussion.

A) We add explanation about this at L253.

Reviewer 2 Report

This study aimed to detect non-responders. This is a significant clinical problem.

However, the author has compared the first to the third injection.

I am not sure this is the correct method. They correlate the first response to the third response.

It would be more clinically relevant to compare the first set of 3 injections to the following one.

This is because clinically- we don’t follow patients after the first injections- this is due to the overcrowding of the clinic- we see the patient once – predictive three injections, then see him again

Doing  a PRN regimen is unobtainable in many centers

Comments:

1 Was these naïve patients?

Two, why not till 2022

Three why >20 years?

4. why not include patients with CME <300

5 How was the treatment allocation? Was it random?

6 why did you choose n=70?

7 r=0.58 why do you consider it “a moderate correlation”? in biological process, r>0.3 is usually observed

8, so it seems from table 2 that if the patient improved after the first injection d7, they would continue to improve dL. Important finding. Also, you could report on % of responders at each visit

9 can you run an analysis to compare those with high d7 – dl to those with low correlation?

We might learn who will respond maybe.

10 So, can you suggest a d7 cut-off value to consider as a non-responder? Would you suggest replacing the drug or continuing to inject it?

Author Response

Reply to Reviewer 2 However, the author has compared the first to the third injection. I am not sure this is the correct method. They correlate the first response to the third response. It would be more clinically relevant to compare the first set of 3 injections to the following one. This is because clinically- we don’t follow patients after the first injections- this is due to the overcrowding of the clinic- we see the patient once – predictive three injections, then see him again Doing a PRN regimen is unobtainable in many centers A) We agree this. So, we add explanation about overcrowding at L82. 1) Was these naïve patients? A) We add ratio of treatment naïve patients at L166. The definition of exclusion criteria was also described at L90. 2) why not till 2022 A) We add explanation about this reason at L82. 3) why >20 years? A) We add explanation about this reason at L84. 4) why not include patients with CME 0.3 is usually observed A) We changed about this at L156 and L195-. 8) so it seems from table 2 that if the patient improved after the first injection d7, they would continue to improve dL. Important finding. Also, you could report on % of responders at each visit. A) We add explanation about this at L181. 9) can you run an analysis to compare those with high d7 – dl to those with low correlation? We might learn who will respond maybe. A) We add this matter as limitation (L270-). 10) So, can you suggest a d7 cut-off value to consider as a non-responder? Would you suggest replacing the drug or continuing to inject it? A) We add ratio of non-responder (CMT>350μm) at L181.
